# Immunization with *Em*CRT-Induced Protective Immunity against *Echinococcus multilocularis* Infection in BALB/c Mice

**DOI:** 10.3390/tropicalmed7100279

**Published:** 2022-10-01

**Authors:** Lujuan Chen, Zhe Cheng, Siqi Xian, Bin Zhan, Zhijian Xu, Yan Yan, Jianfang Chen, Yanhai Wang, Limei Zhao

**Affiliations:** 1Laboratory of Pathogenic Biology, School of Basic Medical Sciences and Forensic Medicine, Baotou Medical College, Baotou 014040, China; 2Parasitology Research Laboratory, School of Life Sciences, Xiamen University, Xiamen 361102, China; 3Department of Pediatrics, National School of Tropical Medicine, Baylor College of Medicine, Houston, TX 77030, USA

**Keywords:** *Echinococcus multilocularis*, alveolar echinococcosis, calreticulin, vaccine, protective immunity

## Abstract

Alveolar echinococcosis (AE) is a severe parasitic zoonosis caused by the larval stage of *Echinococcus multilocularis*. The identification of the antigens eliciting acquired immunity during infection is important for vaccine development against *Echinococcus* infection. Here, we identified that *E. multilocularis* calreticulin (*Em*CRT), a ubiquitous protein with a Ca^2+^-binding ability, could be recognized by the sera of mice infected with *E. multilocularis*. The native *Em*CRT was expressed on the surface of *E. multilocularis* larvae as well as in the secreted products of metacestode vesicles and protoscoleces (PSCs). The coding DNA for *Em*CRT was cloned from the mRNA of the *E. multilocularis* metacestode vesicles and a recombinant *Em*CRT protein (r*Em*CRT) was expressed in *E. coli*. Mice immunized with soluble r*Em*CRT formulated with Freund’s adjuvant (FA) produced a 43.16% larval vesicle weight reduction against the challenge of *E. multilocularis* PSCs compared to those that received the PBS control associated with a high titer of IgG, IgG1 and IgG2a antibody responses as well as high levels of Th1 cytokines (IFN-γ and IL-2) and Th2 cytokines (IL-4, IL-5 and IL-10), produced by splenocytes. Our results suggest that *Em*CRT is an immunodominant protein secreted by *E. multilocularis* larvae and a vaccine candidate that induces partial protective immunity in vaccinated mice against *Echinococcus* infection.

## 1. Introduction

Alveolar echinococcosis (AE) is a severe parasitic zoonosis caused by the larval stage of *Echinococcus multilocularis*, which is rarer but far more severe than cystic echinococcosis (CE) resulting from an infection of *E. granulosus* [1,2]. Humans get infected by the ingestion of infective eggs and the larval stage of the cestode is developed in the liver as cyst-like metacestode vesicles [3]. The vesicles aggressively grow in the liver tissue. Similar to cancer, the vesicles spread within the liver tissue or even metastasize to distant organs, such as the spleen and brain [4]. It is estimated that more than 1 million people are affected with echinococcosis worldwide. Most infected people experience severe clinical syndromes which are life-threatening. The 10-year mortality of untreated AE patients is greater than 90% after diagnosis [5]. Currently, the treatment of AE is mainly through surgery accompanied with anthelminthic chemotherapy [5,6]. However, surgery cannot completely remove the parasite tissue in most cases due to the diffusive and infiltrative feature of the metacestode tissue. Treatment with drugs mainly relies on benzimidazole drugs that usually restrain the growth and dispersion of metacestodes only and cannot eliminate the parasite from an infected liver [6]. These anthelminthic drugs show a limited efficiency against AE, with certain side effects. Therefore, it is necessary to develop a vaccine as an alternative approach to control the infection of *E. multilocularis* which cause AE in endemic areas.

Calreticulin (CRT) is a well-conserved and ubiquitous protein with a Ca^2+^-binding ability among helminths [7]. It mainly exists in the endoplasmic reticulum of all kinds of cells [8], with multifunctions to maintain their parasitism in the hosts [9,10,11]. CRT secreted by *Trichinella spiralis* (TsCRT) binds to the human complement C1q to inhibit the C1q-dependent complement activation and attack the invaded parasite as a survival strategy [12]. A further study identified that the C1q-binding domain was located in the S-domain of the protein and the binding of TsCRT to C1q also inhibited neutrophils to release oxygen species and the formation of neutrophil extracellular traps, so as to inhibit neutrophil-related inflammation [13]; therefore, CRT has been targeted as a vaccine candidate. Immunization with recombinant CRT has induced partial protection against *Schistosoma mansoni* [14], hookworm [15] and *Taenia solium* [16,17]. In this study, we cloned and characterized *Em*CRT, a CRT homologue in *E. multilocularis*, and identified that immunization with bacterial-expressed recombinant *Em*CRT elicited partial protection against *E. multilocularis* infection with Th1/Th2 mixed immune responses in mice. 

## 2. Materials and Methods

### 2.1. Parasite and Animals

Mice used in this study include 6–8-week-old female BALB/c or Kunming mice with weight of 18–21 g or 30–35 g, respectively, purchased from Xiamen University Laboratory Animals Center (XMULAC). All mice were raised at Xiamen University Animal Facility with free access to food and water. The protocols used in this study were prepared in strict accord with good animal practice and approved by Institutional Animal Care and Use Committee of Xiamen University with approval Number: 2013-0053. 

*E. multilocularis* used in this study was originally obtained from an infected fox in Hulunbeier Pasture of Inner Mongolia, China [18], and maintained in Kunming mice as metacestode vesicles as described [19,20,21]. Briefly, the metacestode vesicles were collected from an infected Kunming mouse, then cultivated in DMEM medium supplemented with 10% FBS in a culture flask covered with host feeder Hela cells. The cultivated hydatid tissue was collected and the protoscoleces (PSCs) were isolated by filtering the homogenized hydatid tissue through a 70 μm cell strainer, then washed three times in phosphate-buffered saline (PBS). The collected PSCs were used to infect Kunming or BALB/c mice by intraperitoneal injection of 2000 PSCs per mouse. 

### 2.2. Cloning of Emcrt

The total RNA of *E. multilocularis* was extracted from cultivated metacestode vesicle tissue using RNeasy Mini Kit (Qiagen, Hilden, Germany). The total cDNAs were reverse transcribed from the total RNA using the Evo M-MLV Reverse Transcription Kit with gDNA Clean (ACCURATE BIOLOGY, Changsha, China). The DNA encoding the full-length *Em*CRT without signal peptide was amplified by PCR from the total cDNA using specific primers designed based on the nucleotide sequences of *Em*CRT (UniProtKB accession No. A0A068YA41): the forward primer 5′-ATGGGTCGCGGATCCATGGAAGTTTACTTC-3′ and the reverse primer 5′-GTGGTGGTGCTCGAGCTACAATTCATCCTT-3′. The amplified *Emcrt* cDNA was cloned into the bacterial expression vector pET-28a (Novagen, Darmstadt, Germany) using *Bam*H I and *Xho* I sites. The recombinant *Em*CRT protein (r*Em*CRT) was expressed in *E. coli* BL21 (DE3) under induction of 0.4 mM IPTG at 25 °C for 4 h. The soluble r*Em*CRT with His-tags at N-terminus was purified by Ni-affinity chromatography (Beyotime Biotechnology, Shanghai, China), according to the manufacturer’s instructions. Endotoxin (LPS) was removed by ToxinEraserTM Endotoxin Removal Kit (GenScript Biotech Corporation, Nanjing, China) and measured by the ToxinsensorTM Chromogrnic LAL Endotoxin Assay Kit (GenScript Biotech Corporation, Nanjing, China), according to the manufacturer’s protocol [12]. The purity of r*Em*CRT was analyzed by SDS-PAGE and the antigenicity confirmed by Western blot with anti-*Em*CRT mouse sera and *E. multilocularis* infected mouse sera. The protein concentration was determined using BCA Protein Assay Kit (Beyotime Biotechnology, Shanghai, China). 

### 2.3. Calcium-Binding Staining

The calcium-binding property of r*Em*CRT was assessed by staining with Stains-all (Sigma, St. Louis, MO, USA), a cationic carbo-cyanine dye that stains Ca^2+^-binding proteins as blue and non-calcium-binding proteins as red [22,23]. In the assay, bovine serum albumin (BSA) and the recombinant protein of *Em*Bmi-1 (EmuJ_001132200) were used as non-calcium-binding protein controls.

### 2.4. Expression and Localization of Native EmCRT in E. multilocularis Metacestode Larvae

To determine the mRNA transcription level of *Emcrt* in larval stage of *E. multilocularis*, total RNA was extracted from *E. multilocularis* metacestode vesicles or from isolated PSCs and reversely transcribed into total cDNA as described above, qPCR was performed with HieffTM qPCR SYBR^®^ Green Master Mix (Yeasen Biotechnology (Shanghai) Co., Ltd, China) using *Emcrt* specific primers (forward 5′-TACACGCTCATCATCCGACC-3′ and reverse 5′-TTCCGGTTGTTTGGCATTCG-3′). The housekeeper gene *Emelp* (*E. multilocularis* ERM-like protein, GenBank accession No. AJ012663) was used as an internal control with forward primer 5′-CAGGATCTCTTCGATCAAGTG-3′ and reverse primer 5′-GACCATACTTGGCAACACAG-3′. The results of the threshold cycle (Ct) were calculated using 2^−ΔCt method after being normalized by *Emelp*, and the fold-change of the *Emcrt* gene transcriptional level in *E. multilocularis* PSCs was calculated relative to that in metacestode vesicles.

To determine *Em*CRT protein expression in the different parts of parasite, anti-*Em*CRT serum was generated by immunizing mice with r*Em*CRT as described [12]. PSCs and metacestode vesicles were isolated from infected Kunming mice and cultivated in vitro as described above. Crude somatic extracts of PSCs and metacestode vesicles were obtained by homogenizing the tissue and centrifugation. The vesicle fluid was drawn from inside of the metacestode vesicles and centrifuged at 1000 rpm for 10 min. The fluid supernatant was used as vesicle fluid protein. In addition, the excretory and secretory products (ES) of metacestode vesicles and PSCs were obtained by cultivating PSCs and metacestode vesicles in an RPMI 1640 (Shanghai Basalmedia Technologies Co., LTD, Shanghai, China) without FCS for 48 h at 37 °C with 5% CO_2_. The culture supernatants containing metacestode vesicles and PSCs ES products were concentrated by centrifugation and buffer exchanged into PBS [12]. Afterward, the same amount of all the different tissue samples was separated in 10% polyacrylamide gel, then transferred to PVDF membranes. The *Em*CRT protein was recognized by the anti-r*Em*CRT mouse sera followed by a secondary anti-mouse IgG antibody conjugated with HRP (1:5000 dilution; BOSTER Biological Technology Co. LTD, Chengmai, China). The density of antibody-recognized bands was scanned and semi-quantitatively assessed using the analytic software Image J.

To determine the localization of *Em*CRT in the metacestode of *E. multilocularis*, immunofluorescence assay (IFA) was performed using anti-r*Em*CRT sera as described previously [3]. Briefly, collected PSCs and metacestode vesicles were probed with anti-r*Em*CRT mouse sera (1:1000 dilutions) at 4 °C overnight and then with the secondary antibody Alexa 488-conjugated anti-mouse IgG (Life Technologies, Carlsbad, CA, USA) for 1 h at 37 °C. DNA was counterstained with 4′, 6-diamidino-2-phenylindole (DAPI). Fluorescence images were taken using a Nikon 80i fluorescence microscope (Tokyo, Japan). Contrast and light adjusting and merge of pictures were performed by using Nikon image software supplied by the microscope manufacturer or Adobe Photoshop 8.0 software (San Jose, CA, USA) [24,25,26].

### 2.5. Immunization and Challenge Infection

A total of 39 BALB/c mice were randomly divided into three groups with 13 each. The mice in the first group were each vaccinated subcutaneously with 25 μg of r*Em*CRT emulsified with complete Freund’s adjuvant (FA) [27,28]. The mice were boosted twice with the same amount of protein emulsified with incomplete FA with two weeks interval. Other two groups of mice were injected with PBS + FA or PBS alone as controls. Two weeks after the final boost, 5 mice from each group were euthanized, the sera and spleens were collected for measuring humoral and cellular immune responses. The remaining 8 mice in each group were challenged with 2000 PSCs by intraperitoneal injection in total volume of 100 µL. All challenged mice were sacrificed 14 weeks after infection and the total metacestode vesicles developed in the abdominal cavity were collected to measure the parasite weight [29].

### 2.6. ELISA Measurement of the Antibody Response

Serum was collected from each mouse one week post each immunization. The *Em*CRT-specific IgG and subtype IgG1 and IgG2a were measured in these sera using an indirect enzyme-linked immunosorbent assay (ELISA). Briefly, flat-bottom 96-well plates (Thermo Fisher, Waltham, MA, USA) were coated with 100 μL/well of r*Em*CRT at a concentration of 1.0 μg/mL in bicarbonate buffer (pH 9.6) overnight at 4 °C. After three washes with PBS + 0.05% Tween-20 (PBST), the plates were blocked with blocking buffer (5% BSA in PBS) for 1 h at 37 °C, then probed with serial dilutions of mouse sera for 1 h at 37 °C. After being washed 3 times with PBST, the plates were incubated with HRP-conjugated goat anti-mouse IgG or IgG1 or IgG2a (Invitrogen, Carlsbad, CA, USA) for 1 h at 37 °C. After the final wash, the substrate 3, 3’, 5, 5’-tetramethylbenzidine (TMB, Beyotime Biotechnology, Shanghai,, China) was added to each well, and the reaction was stopped with stop solution (Beyotime Biotechnology, Shanghai,, China). Quantification of the reaction was determined by measuring the absorbance at 450 nm in an ELISA reader [30,31].

### 2.7. Cytokine Analysis

Two weeks after the last immunization, 5 mice from each group were sacrificed and their spleens were collected. The real time qPCR was performed to assess the expression of IL-2 and IFN-γ (Th1), and IL-4, IL-10, IL-5 (Th2) mRNAs in splenocytes collected from different groups of mice using the same method described above by using specific primers designed from their sequences. GAPDH was used as an internal control. The results of the threshold cycle (Ct) were calculated using 2^−ΔCt method after being normalized by GAPDH, the fold-change of the group of r*Em*CRT immunization was calculated relative to that of PBS + adjuvant control or PBS control.

### 2.8. Statistical Analysis

The data were expressed as the means ± standard error. *p* < 0.05 was regarded as statistically significant. Statistical analysis was performed using the Prism 7.0 software (GraphPad Prism software, San Diego, CA, USA). Analysis was carried out using the two-tailed Mann–Whitney test.

## 3. Results

### 3.1. Cloning and Expression of EmCRT 

Based on the sequence analysis, calreticulin is genetically conserved among helminths. The sequence alignment of *Em*CRT with other helminth CRTs reveals that *Em*CRT shares a 98% sequence identity with calreticulin from *E. granulosus* (*Eg*CRT) and an 89% identity with that from cestode *Taenia solium* (TsCRT) (Figure 1A). The phylogenic tree reflects the evolutionary relationships of the organisms involved based on the CRT sequence difference (Figure 1B). We generated a model for the *Em*CRT tertiary structure by SWISS MODEL, which contains three major domains, including the N-domain, internal proline-rich P-domain and C-terminal (Figure 1C). 

The recombinant *Em*CRT protein (r*Em*CRT) was expressed in BL21(DE3) under the induction of 0.4 mM IPTG, and the r*Em*CRT with the 6×His-tag expressed at the N-terminus was purified with immobilized metal affinity chromatography (IMAC). The SDS-PAGE analysis showed that the r*Em*CRT was expressed as a soluble protein with an apparent molecular weight of approximately 58 kDa, which is higher than the predicted size (47 kDa) based on the sequence, possibly due to the high negative charge of *Em*CRT (pI = 4.7) or other structural features [32,33,34,35] (Figure 2A). The purified r*Em*CRT was strongly recognized by the anti-*Em*CRT-antisera raised in the immunized mice as well as by the *E. multilocularis*-infected mouse sera (Figure 2B). The results indicate that the native *Em*CRT can be exposed to the host immune system and induce an antibody response during natural infection. Further staining with the cationic carbocyanine dye “Stains-all” showed r*Em*CRT was stained as dark blue (Figure 2C), indicating it is a Ca^2+^-binding protein [22].

### 3.2. Expression of Native EmCRT in E. multilocularis Metacestode Larvae

The RT-qPCR was performed to determine the transcriptional level of the *Emcrt* mRNA in the *E. multilocularis* PSCs and the total metacestode vesicles. As shown in Figure 3A, we found that the transcriptional level of *Emcrt* in the PSCs was significantly higher (4.1-fold) than that in the total metacestode vesicles (* *p* < 0.05).

The protein expression level of the native *Em*CRT was determined by Western blot, using anti-r*Em*CRT mouse sera. As shown in Figure 3B, one single 58 kDa protein band was clearly identified in the extracts from both the PSCs and metacestode vesicles with a much higher level in the PSCs, which is consistent with the results of the RT-qPCR, indicating *Em*CRT is mostly expressed in PSCs. The *Em*CRT protein was also detected in the metacestode vesicle fluid and the culture supernatants of both the PSCs and metacestode vesicles, indicating *Em*CRT is secreted mostly by PSCs and metacestode vesicles that contain PSCs. 

The IFA with anti-*Em*CRT antisera showed that *Em*CRT was expressed in the germinal layer of the metacestode vesicle (Figure 4A), with a higher level on the surface of the PSCs mainly around the suckers (Figure 4B). In contrast, nothing was shown in both the PSCs and metacestode vesicle when probed with normal mouse sera. Taken together, these results demonstrate that *Em*CRT is constitutively present throughout the *E. multilocularis* larval stages and is especially abundant on the surface of PSCs.

### 3.3. Immune Responses to the Immunization with rEmCRT

The serum samples were gathered from mice one week after each immunization, and the antigen-specific IgG titers against r*Em*CRT were measured using ELISA. The immunization of the mice with r*Em*CRT formulated with FA induced high titers of r*Em*CRT-specific IgG, with the highest titer of 1:512,000 after the third immunization (Figure 5A). The IgG subclass of anti-*Em*CRT antibodies was measured. The results showed that the levels of IgG1 and IgG2a were significantly elevated after the last immunizations, with IgG1 predominant (Figure 5B).

To determine the cellular immune responses upon immunizations with r*Em*CRT, the cytokine responses upon the immunization with r*Em*CRT were measured at the transcription level in splenocytes by a qPCR. Our results showed that the levels of the typical Th1 cytokines (IFN-γ and IL-2) and the Th2 cytokines (IL-4, IL-5 and IL-10) mRNA levels were all increased significantly in the mice vaccinated with r*Em*CRT compared with the mice that received the adjuvant or the PBS control groups, indicating that the r*Em*CRT vaccination induced mixed Th1 and Th2 responses. In addition, the Th2 cytokines IL-4 increased at the highest level, suggesting that r*Em*CRT may favor inducing a Th2-biased immune response (Figure 6), consistent with the result of the IgG subclass antibody levels (with IgG1 dominant over IgG2a).

### 3.4. Partially Protective Immunity Elicited by the Immunization with rEmCRT

The vaccine efficacy of the *Em*CRT immunization was observed by the reduced metacestode vesicle weight in the immunized mice. The result showed that immunization with *Em*CRT exhibited a 43.16% mean reduction in the weight of the total metacestode vesicles developed in the abdominal cavity 14 weeks after the challenge compared to the PBS control group (*p* < 0.05). The adjuvant control group also showed a 21.98% parasite weight reduction compared to the PBS group without significance. These results indicate that immunization with r*Em*CRT induced partial protective immunity against *E. multilocularis* infection in mice (Table 1).

## 4. Discussion

The metacestode stage of *E. multilocularis* is characterized by its tumor-like proliferation and metastasis, causing the life-threatening AE in humans [1,2,4]. The development of effective vaccines for AE has become urgent due to the lack of effective treatment and control means. The identification of the vaccine antigens that induce infective protective immunity and can be manufactured in a large scale is crucial for the vaccine development against AE [36]. In the effort to identify antigens that induce protective immunity against *E. multilocularis* infection, we cloned *Em*CRT from *E. multilocularis* metacestode cDNA and expressed it as a soluble recombinant protein in bacteria. Similar to other CRT, *Em*CRT has a typical structure of three domains, including a globular N-domain, an extended proline-rich P-domain and an acidic C-domain, and also has ability to bind to Ca^2+^ based on the staining with Stains-all [13]. The sequence alignment shows that *Em*CRT shares up to a 50–90% amino acid sequence identity with CRTs from other helminths. Importantly, *Em*CRT shares up to a 98% sequence identity with its counterpart in the closely related *E. granulosus* and an 89% identity with cestode *T. solium*, indicating immunization with *Em*CRT may produce cross-protective immunity to other cestode infections if *Em*CRT is a good vaccine candidate against *E. multilocularis* infection. The Western blot with anti-*Em*CRT mouse sera identified *Em*CRT was dominantly expressed in the PSCs of *E. multilocularis* and less in the metacestode vesicles, possibly because the vesicles contain some PSCs inside as well. Significantly, *Em*CRT was observed in the vesicle fluid and ES products of cultured PSCs and vesicles. IFA staining showed its expression on the germinal layer of the vesicles and more around the sucker of the scolex, indicating native *Em*CRT is secreted in the host environment and possibly involved in the protection of the invading parasite as a survival strategy. TsCRT from *Trichinella spiralis* has been identified as binding to the human complement component C1q and inhibiting the C1q-involved complement attack and neutrophil reaction to the infection of the nematode. However, the function of *Em*CRT is not clear yet in the role of *E. multilocularis* parasitism in the host.

Due to its secretory property and the potential functions involved in the survival of the parasitic helminth within the infected host, the recombinant *Em*CRT expressed in *E. coli* was applied to immunize mice for testing its immunogenicity and vaccine efficacy against the challenge of *E. multilocularis* PSCs. The results of the vaccine trial with *Em*CRT demonstrated that mice immunized with *Em*CRT formulated with Freund’s adjuvant produced a 43.16% reduction in the metacestode vesicles weight with statistical significance compared to mice that received the PBS only. It also showed a 21.98% reduction in the metacestode vesicle weight in the group that received the adjuvant only compared to the PBS control even though it is not statistically significant; possibly, the adjuvant itself boosts the mouse immune response that induces a certain resistance. The metacestode vesicle reduction in *Em*CRT-immunized mice is associated with significantly high titers of the IgG response. An IgG subtype analysis revealed IgG1 was dominant over IgG2a, indicating a Th2-biased immune response. The cytokine profile based on the mRNA expression showed that immunization with *Em*CRT induced splenocytes to produce both Th1 (IFN-γ and IL-2) and Th2 cytokines (IL-4, IL-5 and IL-10). However, the IL-4 was mostly induced to the highest level, combining with the dominant IgG1 response over IgG2a, and it indicates that the immunization of *Em*CRT formulated with FA induced both Th1 and Th2 responses, with Th2 dominant [37,38,39]. In general, our data indicated that *Em*CRT is a highly immunogenic antigen and induces a strong antibody response and Th1/2 cytokine response in immunized mice that may contribute to the protective immunity against the *E. multilocularis* infection in this study. The protective immunity was also observed in the immunization of CRTs from other helminths. More than one-third of hamsters infected with *T. solium* elicited anti-TsCRT IgG antibodies and IL-10 production [17]. Hamsters immunized with TsCRT produced 40–100% protection against *T. solium* oral infection, depending on the type of infected cysticerci [16]. Mice immunized with hookworm calreticulin intraperitoneally without adjuvant produced 43–49% fewer worms in their lungs following a hookworm larva challenge associated with low levels of serum IgE and moderate lung eosinophilia [15]. The spleen T cells from mice immunized with irradiated *S. mansoni* cercariae and acquired protection against *S. mansoni* infection strongly reacted to SmCRT with IL-4 production [14,39], indicating that CRT is able to induce a Th2 response and protective immune effect during helminth infections [40]. This evidence, combined with our observation in this study, further supports that *Em*CRT is a good vaccine candidate against *E. multilocularis* infection. However, the *Em*CRT immunization-induced metacestode vesicle weight reduction in this study was not high enough (43.16% compared to the PBS control). Further research is needed to optimize the protective immunity, such as the immunization route, different adjuvants or dosage, in order to achieve higher protection against *E. multiocularis* infection. Due to the complex life cycle and diversity of antigens in different stages, it may be necessary to use the combination of multiple antigens from different developmental stages or a multiepitope vaccine combined with T- and B-cell epitopes from different protective antigens [41] to increase protection against parasitic helminth infections. Some T-cell and B-cell epitopes have been identified from *E. multiocularis* based on the bioinformatic analysis [42], which is under investigation for a vaccine trial against AE. 

In conclusion, this work demonstrates that *Em*CRT plays an important role in inducing partially protective immunity in the immunized mice and therefore could be considered a potential candidate for vaccine development against AE. It is necessary to optimize the protective immunity by changing the immunization route or regime or combining multiple effective vaccines to improve the vaccine efficacy against alveolar echinococcosis. 

## Figures and Tables

**Figure 1 tropicalmed-07-00279-f001:**
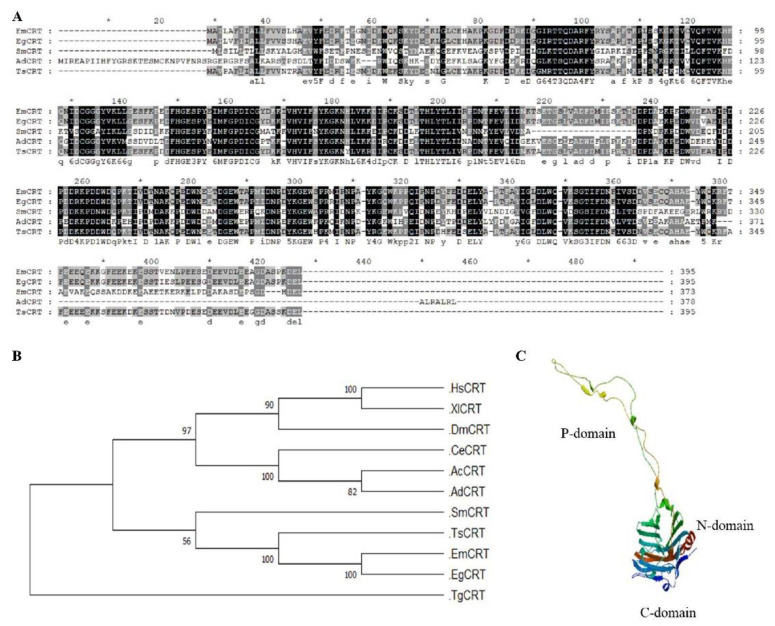
Amino acid sequence analysis and predicted three-dimensional structure of *Em*CRT. (**A**) Amino acid sequence alignment of *Em*CRT with its homologs from *Echinococcus granulosus* (EgCRT), *Schistosoma mansoni* (SmCRT), *Taenia solium* (TsCRT) and *Ancylostoma duodenale* (AdCRT) using the GeneDoc software. Numbers on the right refer to the last amino acid in each corresponding organism. Residues with dark shading indicate the same amino acids. Grey shading represents 80–90% similarity and light grey represents 60–70% similarity. (**B**) Phylogenetic analysis of the *Em*CRT and its homologues. Phylogenetic tree was generated with MEGA 6.0 by the neighbor-joining method (bootstrap = 1000). The numbers on the branches represent bootstrap values. *Homo sapiens* (HsCRT), *Xenopus laevis* (XlCRT), *Drosophila melanogaster* (DmCRT), *Caenorhabditis elegans* (CeCRT), *Angiostrongylus cantonensis* (AcCRT) and *Trypanosoma grayi* (TgCRT). (**C**) Modeled 3D structure of *Em*CRT.

**Figure 2 tropicalmed-07-00279-f002:**
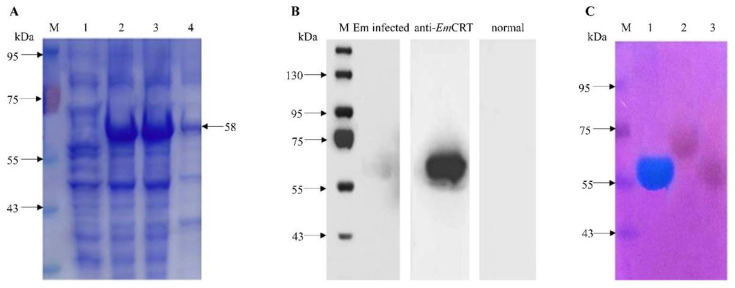
Expression and characterization of r*Em*CRT. (**A**) Expression of r*Em*CRT in *E. coli* BL21 as soluble protein. After being induced with 0.4 mM IPTG for 4 h, the soluble fraction of induced (lane 3) or the whole induced lysate (lane 2) compared with lysate without induction (lane 1). Lane 4, purified r*Em*CRT (2 ug). (**B**) Recognition of r*Em*CRT by specific antibodies. Western blot analysis of r*Em*CRT (500 ng) with *E. multilocularis*-infected mouse sera, anti-r*Em*CRT mouse sera and with normal mouse sera. (**C**) Purified r*Em*CRT and other control proteins (100 ug each) stained with Stains-all. Lane 1, r*Em*CRT stained in blue; Lane 2, r*Em*Bmi-1-GST as a *E. multilocularis* irrelevant protein control; and Lane 3, BSA control stained in red. M, molecular weight marker.

**Figure 3 tropicalmed-07-00279-f003:**
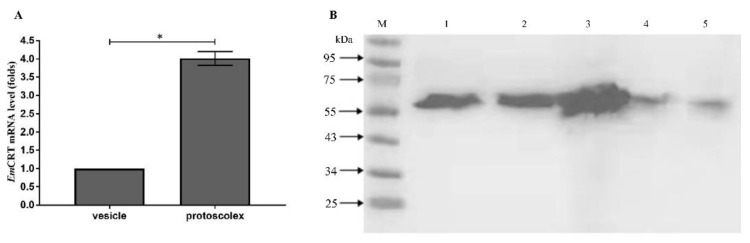
Expression of *Em*CRT in *E. multilocularis* metacestode larvae. (**A**) RT-qPCR analysis of transcription level of *Emcrt* mRNA in PSCs and metacestode vesicle tissue. After being normalized with *Emelp*, the fold-change of the *Emcrt* mRNA transcription in PSCs was calculated relative to that in metacestode vesicles. * *p* < 0.05. (**B**) Analysis of *Em*CRT protein expression in the lysates of metacestode vesicles (Lane 1), metacestode vesicle fluid (Lane 2), PSCs (Lane 3) and the excretory–secretory products released by cultured metacestode vesicles (Lane 4) and by PSCs (Lane 5) by Western blot with anti-r*Em*CRT mouse sera. Equal amount of total protein (100 ug), determined by Bradford assay, was loaded for each sample.

**Figure 4 tropicalmed-07-00279-f004:**
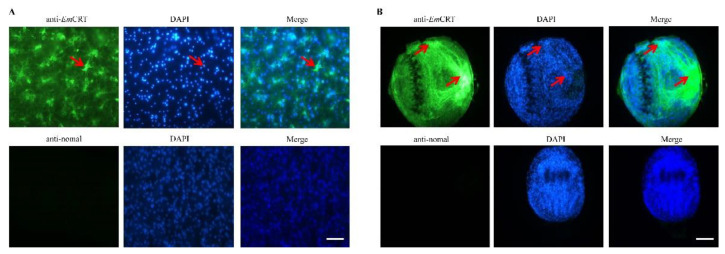
Immunolocalization of *Em*CRT in *E. multilocularis* metacestode larvae. Metacestode vesicles (**A**) and PSCs (**B**) were incubated with anti-*Em*CRT mouse serum and normal mouse serum, respectively, then probed with Alexa 488-conjugated anti-mouse IgG. The nuclei were stained with 4’, 6-diamidino-2-phenylindole (DAPI, blue). Scale bar = 100 μm. The arrows indicate the area with highly expressed *Em*CRT.

**Figure 5 tropicalmed-07-00279-f005:**
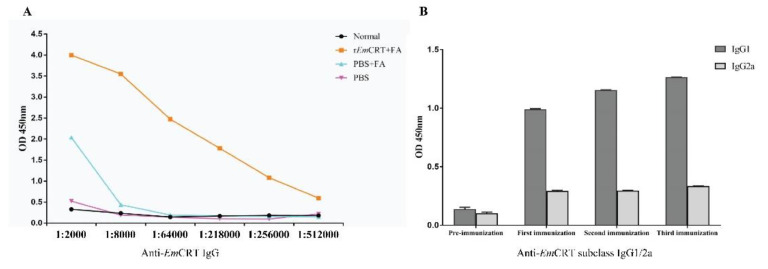
Antibody responses to the immunization with r*Em*CRT. (**A**) Total anti-*Em*CRT IgG titers in sera of all groups mice after last immunization. (**B**) OD450 value of anti-*Em*CRT IgG1 and IgG2a in sera of immunized mice with dilution 1:200. The values are presented as the arithmetic mean of five mice in the r*Em*CRT group ± standard error.

**Figure 6 tropicalmed-07-00279-f006:**
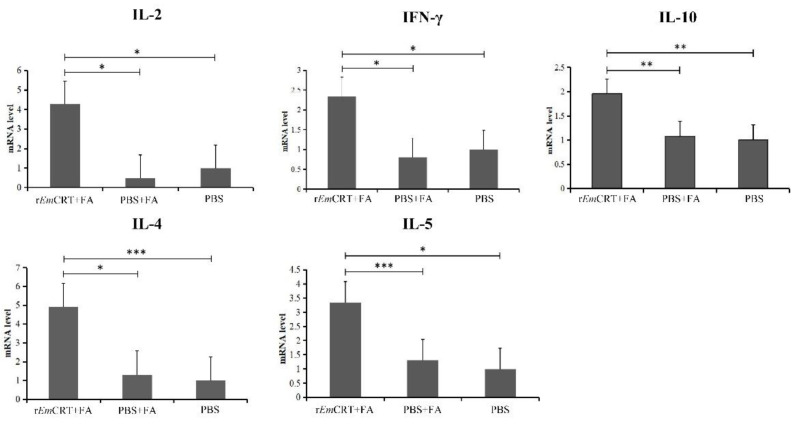
Splenocyte cytokine transcriptional profile of mice immunized with r*Em*CRT. Total cDNA was prepared from total RNA obtained from spleens of mice immunized with r*Em*CRT three times. The mRNA levels of IL-2, IFN-γ, IL-4, IL-5 and IL-10 were determined by RT-qPCR. Data are expressed as the mean of the ratio of each cytokine relative to GAPDH (housekeeping gene). * *p* < 0.05, ** *p* < 0.01, *** *p* < 0.001.

**Table 1 tropicalmed-07-00279-t001:** Alveolar echinococcus weight in mice.

Groups	Metacestode Vesicle Weight	Mean Reduction in Parasite Weight (Compared with PBS)
r*Em*CRT + FA	0.8539125 ± 0.337201921	43.16% *
PBS + FA	1.1720375 ± 0.494765754	21.98%
PBS	1.50224 ± 0.69753474	

Metacestode vesicle weight in each group of mice after challenge with 2000 protoscoleces. Metacestode vesicle weight is presented as the mean ± S.D. The asterisks indicate statistically significant differences in metacestode vesicle weight compared to the PBS group (* *p* < 0.05).

## Data Availability

Data included in this article are available from the corresponding author upon request and are also available at preprint Research Squire with DOI: https://doi.org/10.21203/rs.3.rs-1740725/v1 accessed on 26 September 2022.

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
