# Peer review of "Immunization with EmCRT-Induced Protective Immunity against Echinococcus multilocularis Infection in BALB/c Mice"

_tropicalmed, 2022, doi:10.3390/tropicalmed7100279_

Round 1

Reviewer 1 Report

General comments:

Alveolar echinococcosis is a severe parasitic zoonosis. Therefore, it is necessary to develop vaccine as an alternative approach to control infection of E. multilocularis and caused AE in endemic areas.

In this study, researchers aimed to investigate whether E. multilocularis calreticulin (EmCRT) secreted by larvae of E. multilocularis can be a vaccine candidate against AE.

Although different routes of immunization, adjuvants and dosing have not been tried in the study, this work demonstrates that EmCRT plays an important role in inducing partially protective immunity in the immunized mice and therefore could be considered a potential candidate for vaccine development against AE. Also, it presents immunization of EmCRT may produce cross-protective immunity to other cestode infections.

Manuscript is well designed and written. The study result is important as it is promising against the control of AE.

Specific comments:

P.2. L.49: …….liver [6]. These anthelminthic drugs….

P.2. L.83: …….times in phosphate-buffered saline (PBS). The collected……

P.2. L.85: As I understand, per mouse were infected with 2000 PSCs by intraperitoneal injection for challenge. If so, please remove the cyst fluids form here (2000 PSCs or 0.2 ml hydatid fluid per mouse).

P.2. L.95: Please, add related reference to primers.

P.3. L.106: …. Assay Kit (Beyotime, add brand, add country). Please, correct it and all throughout the entire the manuscript.

P.3.L.122: 5′-………..CAG-3′).

P.3.L.136: …….into PBS (12).

P.3.L.141: China).The…

Author Response

Point 1: P.2. L.49: …….liver [6]. These anthelminthic drugs….

Response: Thanks. It has been amended in the revised manuscript. (see line 48)

Point 2: P.2. L.83: …….times in phosphate-buffered saline (PBS). The collected……

Response: Thanks. It has been amended in the revised manuscript. (see line 82)

Point 3: P.2. L.85: As I understand, per mouse were infected with 2000 PSCs by intraperitoneal injection for challenge. If so, please remove the cyst fluids form here (2000 PSCs or 0.2 ml hydatid fluid per mouse).

Response: Thanks. 0.2 ml hydatid fluid has been deleted in the revised manuscript. (see line 82-83)

Point 4: P.2. L.95: Please, add related reference to primers.

Response: Thanks. The primers were designed based on the actual sequence of  A0A068YA41, no reference cited.

Point 5: P.3. L.106: …. Assay Kit (Beyotime, add brand, add country). Please, correct it and all throughout the entire the manuscript.

Response: Thanks. We added countries in all reagents through the manuscript.

Point 6: P.3.L.122: 5′-………..CAG-3′).

Response: Thanks. It has been amended in the revised manuscript. (see line 119)

Point 7: P.3.L.136: …….into PBS (12).

Response: Thanks. It has been amended in the revised manuscript. (see line 134)

Point 8: P.3.L.141: China).The…

Response: Thanks. It has been amended in the revised manuscript. (see line 138)

Reviewer 2 Report

English language needs to be checked.

Author Response

Point 1: English language needs to be checked.

Response: Thank you for your suggestion. The revised manuscript has been reviewed by an editor with English as first language and all grammatical errors and typos we found have been corrected.

Reviewer 3 Report

The current manuscript explores the possible suitability of the protein calreticulin of Echinococcus multilocularis as a potential vaccine candidate for alveolar echinococcosis The authors have done a great job with the amount of effort concerning the appropriate study design and a rather clearly written manuscript. Made for an interesting read.

I do only have minor comments since the manuscript appears to be at a rather polished stage:

Introduction:

A minor detail, but humans are not intermediate hosts for E. multilocularis – for a host to be considered as an intermediate host it would require that another host would eat the infected organs. Humans are dead-end accidental hosts, infected with the larval stage. Please do amend.

Material and methods:

L80-85: Please note that according to recent international consensus (Vuitton et al. 2020) it was recommended to avoid using the term “hydatid fluid/cyst”, emphasizing that “hydatid fluid/cyst” should only be used to describe CE. For E. multilocularis the appropriate term would be “vesicle fluid” from in vitro metacestode vesicles. Please amend.

Results:

L215-222: More appropriate as a section in material and methods. No need to re-iterate the methods in detail in the results section.

Figure 4: the legend to figure needs to be more detailed and also describe the abbreviations (e.g., DAPI). Furthermore, the legend states that there should be an arrow to indicate the area with highly expressed EmCRT – there is no arrow.

Discussion:

L353: I would suggest avoiding using “worm” to describe the metacestode stage of E. multilocularis. It is inaccurate. Please amend throughout the discussion.

Author Response

Point 1: introduction: 

A minor detail, but humans are not intermediate hosts for E. multilocularis – for a host to be considered as an intermediate host it would require that another host would eat the infected organs. Humans are dead-end accidental hosts, infected with the larval stage. Please do amend.

Response: Thank you for your correction, “As its intermediate host” (Line 36) has been deleted.

Point 2: Material and methods: 

L80-85: Please note that according to recent international consensus (Vuitton et al. 2020) it was recommended to avoid using the term “hydatid fluid/cyst”, emphasizing that “hydatid fluid/cyst” should only be used to describe CE. For E. multilocularis the appropriate term would be “vesicle fluid” from in vitro metacestode vesicles. Please amend.

Response: Thank you for your correction. We have checked the full text thoroughly and corrected hydatid to vesicle fluid in the revised manuscript.

Point 3: Results:

L215-222: More appropriate as a section in material and methods. No need to re-iterate the methods in detail in the results section.

Response: Thanks. The redundant method description has been deleted in the revised manuscript. (see line 213-214)

Figure 4: the legend to figure needs to be more detailed and also describe the abbreviations (e.g., DAPI). Furthermore, the legend states that there should be an arrow to indicate the area with highly expressed EmCRT – there is no arrow.

Response: Thanks. Figure legend has been amended in the revised manuscript based on your suggestions (see line 265-270)

Point 4: Discussion:

L353: I would suggest avoiding using “worm” to describe the metacestode stage of E. multilocularis. It is inaccurate. Please amend throughout the discussion.

Response: Thanks. We have checked the discussion thoroughly and remove “worm” from the revised manuscript.